# Long Noncoding RNAs and Circular RNAs in Autoimmune Diseases

**DOI:** 10.3390/biom10071044

**Published:** 2020-07-14

**Authors:** Valeria Lodde, Giampaolo Murgia, Elena Rita Simula, Maristella Steri, Matteo Floris, Maria Laura Idda

**Affiliations:** 1Department of Biomedical Sciences, University of Sassari, Viale San Pietro 43/b, 07100 Sassari, Italy; valerialodde@gmail.com (V.L.); gmurgia22@gmail.com (G.M.); simulaelena@gmail.com (E.R.S.); matteo.floris@gmail.com (M.F.); 2Istituto di Ricerca Genetica e Biomedica, Consiglio Nazionale delle Ricerche, SS554 km 4,500, 09042 Monserrato-Cagliari, Italy; maristella.steri@irgb.cnr.it; 3Istituto di Ricerca Genetica e Biomedica, Consiglio Nazionale delle Ricerche, Traversa La Crucca 3, 07100 Sassari, Italy

**Keywords:** post-transcriptional gene regulation, autoimmunity, lncRNA, circRNA, immune system

## Abstract

Immune responses are essential for the clearance of pathogens and the repair of injured tissues; however, if these responses are not properly controlled, autoimmune diseases can occur. Autoimmune diseases (ADs) are a family of disorders characterized by the body’s immune response being directed against its own tissues, with consequent chronic inflammation and tissue damage. Despite enormous efforts to identify new drug targets and develop new therapies to prevent and ameliorate AD symptoms, no definitive solutions are available today. Additionally, while substantial progress has been made in drug development for some ADs, most treatments only ameliorate symptoms and, in general, ADs are still incurable. Hundreds of genetic loci have been identified and associated with ADs by genome-wide association studies. However, the whole list of molecular factors that contribute to AD pathogenesis is still unknown. Noncoding (nc)RNAs, such as microRNAs, circular (circ)RNAs, and long noncoding (lnc)RNAs, regulate gene expression at different levels in various diseases, including ADs, and serve as potential drug targets as well as biomarkers for disease progression and response to therapy. In this review, we will focus on the potential roles and genetic regulation of ncRNA in four autoimmune diseases—systemic lupus erythematosus, rheumatoid arthritis, multiple sclerosis, and type 1 diabetes mellitus.

## 1. Introduction

While immune responses are essential for efficient defense and clearance against pathogens, their dysregulation can initiate chronic reactions that, when unsolved, can lead to autoimmune disorders. Autoimmune diseases (ADs) are chronic disabling diseases originating from the loss of immunologic tolerance to self-antigens, abnormal activation of the immune-mediated inflammation, and damage of target organs [1]. ADs are multifactorial diseases and host genetic factors, aberrant immune responses, and environmental factors can all contribute to their development [2].

Currently, there are over 100 types of ADs that affect 5–10% of the population worldwide with enormous health care costs due to high rates of comorbidity and multi-organ involvement [3]. Common ADs include multiple sclerosis (MS), psoriasis, type 1 diabetes mellitus (T1D), Graves’ disease, inflammatory bowel disease, rheumatoid arthritis (RA), systemic lupus erythematosus (SLE), and Sjögren’s syndrome [4]. The early symptoms of many ADs are very similar, and include fatigue, achy muscles, swelling and redness, low-grade fever, trouble concentrating, numbness and tingling in the hands and feet, hair loss, and skin rashes [5]. From a pathological point of view, these disorders are characterized by the production of autoantibodies [6,7,8,9] and abnormalities in T and B cell signaling, cytokine secretion, and chemokine production [10,11,12,13]. However, each autoimmune disease is characterized by specific features that we analyze below in this manuscript.

Noncoding RNAs are transcribed from DNA but not translated into proteins [14]. They can bind to DNA, RNA, and proteins to regulate various cellular processes, including gene transcription, RNA turnover, mRNA translation, and protein assembly [14]. Functionally, they can be divided into housekeeping or structural noncoding RNAs (ubiquitously expressed, e.g., transfer RNAs and ribosomal RNAs), and regulatory ncRNAs [15]. Regulatory ncRNAs, such as small interfering RNAs (siRNAs), micro-RNAs (miRNAs), piwi-RNAs (piRNAs), and long ncRNAs (lncRNAs) are expressed in specific cell types in response to developmental signals and environmental stimuli [14,15]. They are key regulators of multiple protein-coding genes implicated in physiological and pathological processes [16]. Additionally, ncRNAs can act as molecular sponges, sequestering other ncRNAs and RNA binding proteins, thus inhibiting the interactions of these molecules with their target RNAs. The crosstalk of these molecular connections enables the fine-tuned regulation of a wide range of biological processes, while the abnormal expression of ncRNAs interferes with mRNA expression patterns creating a dysregulation that can culminate in disease development [17,18,19]. Regulatory lncRNAs (the focus of this review), including circular (circRNAs), have been implicated in AD pathogenesis [20] and may serve as biomarkers and new drug targets for the identification of alternative therapies [21,22,23].

LncRNAs are a class of ncRNAs that range from ~200 bases to hundreds of kilobases [24]. LncRNAs can be transcribed from different genomic regions such as exons, promoters, and intergenic thus generating different classes of lncRNA described in Figure 1 [25].

LncRNAs control gene expression of target genes through actions at the transcriptional, post-transcriptional, and post-translational levels [24,26,27]. LncRNAs can also function as sponges for molecules such as miRNAs and RBPs (RNA binding proteins), thereby influencing their availability for other molecular processes [28]. LncRNAs are expressed and involved in the regulation of several molecules and cells of the immune system comprising T and B lymphocytes, macrophages, monocytes, dendritic cells, and NK cells [29], thus, dysregulation of lncRNAs is associated with autoimmunity onset [30]. Recently, several lines of evidence indicated that lncRNAs can circularize originating circRNAs [31]. As an important member of the lncRNAs, circRNAs have drawn great attention in recent decades. Unlike linear RNA, circRNAs are a covalently-linked single-stranded RNA without 5′ and 3′ ends [32]. To date, many circRNAs have been identified, with the majority of them having the following characteristics—they are abundant, conserved, stable, and specific [33,34]. Based on their structures, we can identify three main classes of circRNAs—exon–intron circular RNAs (EIciRNAs), intronic circular RNAs (ciRNAs), and exonic circular RNAs (ecircRNAs) (Figure 2). 

CircRNAs are involved in the pathogenesis of several human diseases and have a range of functions, including functioning as miRNA and RBP sponge, RNAP II elongation, RNA maturation regulation through regulation of alternative splicing, protein localization, and translation [35]. Recently, alterations in the expression of many circRNAs have been reported to play important roles in the onset and progression of autoimmune diseases [35]. Thus, circRNAs may serve as potential biomarkers and act as immune regulators offering potential opportunities for the identification of new therapies [36]. 

In this review, we provide an overview of the latest studies analyzing the role of lncRNAs in the contest of four human ADs: RA, SLE, MS, and T1D, (Table 1 and Table 2). We also highlight the relevance of these molecules as biomarkers and the role of genetic variability in their regulation.

## 2. Role of ncRNAs in Rheumatoid Arthritis

### 2.1. Rheumatoid Arthritis

RA is a systemic autoimmune disease initially affecting the lining of the synovial joints [67]. It induces cartilage and bone damage, leading eventually to progressive disability and premature death [67]. From a pathological point of view, RA is characterized by chronic inflammation as well as immune-mediated cartilage and subchondral bone damage that leads to joint deformities [68]. 

The detection of autoantibodies in asymptomatic subjects is possible up to 10 years before the clinical manifestation of the disease, thus suggesting that the loss of tolerance to autoantigens occurs long before the clinical disease onset [69]. Anti-citrullinated protein/peptide antibodies (ACPA) are the most specific biological markers with predictive and prognostic value in RA patients: they increase in 67% of patient sera five to ten years prior to diagnosis of RA [70]. Early diagnosis is considered a key factor for the most desired outcomes but until now it remains challenging because early symptoms and signs imitate those of several other diseases [71,72].

Complex mechanisms regulating the crosstalk between innate and adaptive immune cells activate the onset and maintenance of synovial tissue inflammation [73]. Leukocytes infiltrate the synovial compartment, and the synovial fluid is flooded with pro-inflammatory mediators such as IL-1β, TNF-α, and IL-6 [74,75]. These events lead to the activation of an inflammatory cascade, characterized by interactions of fibroblast-like synoviocytes with the cells of both the innate and adaptive immune system, including macrophages, T cells, and B cells [74,75].

Recently, research on ncRNAs has unfolded a key role of these molecules in the regulation of the innate and the adaptive immune system in RA pathogenesis [76]. Additionally, genetic variance of some ncRNAs encoding genes might predispose an individual to RA development and identifying alteration of these molecules is a valid diagnostic and prognostic element. Today several noncoding RNAs have been identified and validated as promising biomarkers for the diagnosis and treatment of RA.

### 2.2. LncRNA and Rheumatoid Arthritis

Recently, several studies demonstrated that dysregulation of lncRNAs plays a critical role in the pathogenesis of RA. Today the most used molecular approaches to identify RA-related lncRNAs are microarray analysis and whole transcriptome and quantitative real-time PCR, and the targeted cells for these studies range from peripheral blood cells to fibroblast like synoviocytes (FLSs). Using different approaches, several studies demonstrated that lncRNAs are differentially expressed in RA patients as compared with healthy controls, and this differential expression can be associated with disease features.

Yuan and colleagues, for example, analyzed the peripheral blood mononuclear cells (PBMCs) and identified 2099 lncRNAs and 2307 mRNAs differentially expressed between RA patients and healthy controls. Among them, the lncRNA AC000061, an antisense RNA transcribed from the cystic fibrosis transmembrane conductance regulator (CFTR) gene, was positively correlated with the serum levels of IL-6 and TNF-α as well as the Simplified Disease Activity Index (SDAI) of the RA patients [37]. Furthermore, Zhang et al. used microarrays to profile the alterations of lncRNAs in FLSs of tissues isolated from knee joints of RA patients and heathy controls. They identified 135 lncRNAs that were differentially expressed between the two groups [77]. Among them, dysregulation of the lncRNAs ENST00000483588, ENST00000438399, uc004afb.1, and ENST00000452247 were validated by RT-qPCR. Furthermore, functional studies and co-expression network analysis suggested a possible involvement of these lncRNA in the pathological processes of RA and that these lncRNAs may have potential value as biomarkers for RA [77].

One of the first lncRNAs characterized for the pathogenesis of RA was HOX transcript antisense RNA (HOTAIR), initially reported as a player in cancer pathogenesis [38]. Song et al. reported that expression of HOTAIR is upregulated in PBMCs and serum exosomes of patients with RA compared to healthy controls. Upregulated HOTAIR can promote the migration of active macrophages. Furthermore, a downregulation of HOTAIR detected in differentiated osteoclasts and rheumatoid synoviocytes correlated with decreased levels of matrix metalloproteinases (MMPs) such as MMP-2 and MMP-13 [39]. MMPs are important players in survival, proliferation, migration, and invasion of RA synovial fibroblasts [78]. These results indicate that aberrant expression of HOTAIR is involved in the pathogenesis of RA and that HOTAIR could be used as potential biomarker for RA diagnosis [39].

As mentioned above, overexpression of lncRNAs can also be detected in T cells and B cells, and synovial tissues of patients with RA. Unfortunately, the roles of these transcripts in FLSs are still unknown. To unravel this point, Mo and colleagues compared the expression of the long intergenic noncoding (linc)RNA gastric adenocarcinoma-associated, a positive CD44 regulator and long intergenic non-coding RNA (GAPLINC), a positive CD44 regulator, in FLSs obtained from RA patients with those derived from patients suffering from common traumatic injury. They observed increased GAPLINC expression in FLSs from RA patients (RA-FLSs) as compared with patients with traumatic injury. Moreover, GAPLINC suppression in RA-FLS cells, has been associated with a significant decrease in cell proliferation, invasion, migration, and production of proinflammatory cytokines such as IL-6, IL-8, and MMP-9. Additionally, the silencing of GAPLINC led to the increased expression of miRNAs, miR-382-5p and miR-575. Mo and colleagues suggested that GAPLINC may function as a sponge for miRNAs thereby affecting the biological characteristics of RA-FLSs and contributing to the development of RA [28].

Upregulation of the lncRNA ZNFX1 antisense RNA 1 (ZFAS1) has been initially observed in cancers promoting cell migration and invasion. In 2018, Ye and colleagues analyzed the expression and function of ZFAS1 in RA patients. They found that the expression of ZFAS1 is upregulated in synovial tissue and FLS cells purified from RA patients as compared with healthy patients. Furthermore, using in vitro assays they demonstrated that ZFAS1 silencing reduced the migration and invasion of RA-FLS. Ye and colleague, finally, showed that ZFAS1 interacted with the miRNA-27a decreasing its functionality. Overall, the results indicated that ZFAS1 promotes RA-FLS migration and invasion in an miR-27a-dependent manner, suggesting a role of this lncRNA as a therapeutic target for RA patients [40]. Another lncRNA with altered expression in RA is growth-arrest-specific 5 (GAS5) which acts as a potent repressor of the glucocorticoid receptor [79]. Mayama and colleagues examined the expression of GAS5 in different immune cell populations related to RA diseases. GAS5 was significantly reduced in the CD4+ T cells and B cells purified from patients with RA as compared with healthy controls [41]. These results are fundamental to expanding information on the role of lncRNAs in RA and providing both new targets for innovative research and new biomarkers and drug targets.

### 2.3. CircRNA and Rheumatoid Arthritis

The role of circRNAs in RA has not been broadly explored and the role in RA pathogenesis is still unclear. However, the expression profiles of circRNAs have been analyzed in different studies. For example, using total RNA from PBMCs of 10 RA patients and 10 healthy controls, circRNA expression profiling was analyzed by microarray analysis. The study, performed by Zheng and colleagues, identified ~ 600 circRNAs differentially expressed in PBMCs of RA donors as compared with healthy controls [55]. Validation using RT-qPCR confirmed differential expression of the circRNA_104194, circRNA_104593, circRNA_103334, circRNA_101407, and circRNA_102594. They also found that two circRNAs (circ_0092285 and circ_0058794) were more expressed while two other circRNAs (circ_0088088 and circ_0038644) were under-expressed in RA. circ_0038644 is encoded by the PRKCB (protein kinase C beta) gene which is associated with the lipopolysaccharide (LPS) immune response and affects the viability of B cells and the antigenic response [55]. Furthermore, the expression of the circRNAs, circ_0000175 and circ_0008410, was analyzed in PBMCs of 24 patients with RA and 24 healthy controls using RT-qPCR analysis by Luo ad colleagues. The levels of circ_0000175 and circ_0008410 correlated with anti-citrullinated protein antibodies and tender joint count, respectively, suggesting their expression may improve the diagnostic accuracy for RA. Thus, the expression levels of the circ_0000175 and circ_0008410 correlate with disease activity and severity [56]. These results corroborate the role that circRNAs may exhibit in the regulation of expression of target pathways that influence the occurrence and development of RA.

Interestingly, many studies have shown that the circRNAs can interfere with miRNA functions, behaving as “miRNA sponges”. miR-16, for example, plays an important role in RA and is more expressed in RA patients. The level of miR-16-5p, which is one of the circ_0088088 targets, correlated with the Th17/Treg cell imbalance and the degree of disease activity, such as C-reactive protein (CRP) levels and disease activity score (DAS28) [55]. ciRS-7 was also upregulated in RA peripheral blood compared with healthy controls and positively correlated with the level of anti-CCP [80]. These studies suggest that specific circRNAs may be diagnostic RA markers or potential therapeutic targets.

## 3. Role of ncRNAs in Systemic Lupus Erythematosus

### 3.1. Systemic Lupus Erythematosus

SLE is a complex, chronic, and systemic autoimmune disorder characterized by a heterogeneous spectrum of clinical and serological signs generated mainly by the production of autoantibody, the activation of the complement cascade, and the deposition of immune complex [81,82]. A remarkable feature of SLE is the production of antinuclear antibodies (ANAs) targeting double-stranded DNA (dsDNA) and other nuclear autoantigens originating from uncleared apoptotic cells [83].

SLE can affect nearly every organ and tissue, resulting in manifestations including fever, rashes, arthritis, vasculitis, and life-threatening nephritis [84]. Due to the high morbidity and mortality, it is fundamental to identify the mechanisms leading to SLE onset, and to isolate reliable biomarkers for early diagnosis, prognosis, and effective therapy. The development of SLE involves practically the entire immune system [85] with complex interactions of genetics and environmental factors that lead to immune dysregulation and diminished tolerance to self-antigens. Lymphocytes were the first cells considered in SLE pathogenesis as aberrantly-activated T cells mediate inflammatory responses which activate B cells to differentiate and produce more autoantibodies, contributing to disease development [86,87]. The role of plasmacytoid dendritic cells (pDCs) has attracted the attention of scientists in recent years, as pDC infiltrations in the renal and skin lesions of SLE patients were identified long ago [88]. pDCs produce a high amount of type I interferon (IFN), a cytokine which is generally elevated in SLE patients [89]. IFNs in turn, enhance the immune responses by modulating T cell and B cell functions [90]. Today the diagnosis of SLE is primarily clinical and remains challenging because of the heterogeneity of the pathology [91]

In recent years, the cellular and molecular roles of noncoding RNAs (miRNA, lncRNA, and circRNA) have been reported in SLE pathogenesis. Indeed, a better understanding of the roles of these molecules is emerging as fundamental in the diverse biological processes leading to SLE including alterations of immune cells and key molecular mechanisms [92,93]. Circulating ncRNAs have also been identified as biomarkers for SLE diagnosis, progression, and activity, demonstrating they play important roles in both SLE pathogenesis and as diagnostic tools for SLE [94].

### 3.2. LncRNA and Systemic Lupus Erythematosus

SLE patients generally present high heterogeneity in pathogenesis and disease features which makes it difficult to understand the etiology of the disease. Indeed, while the pathogenesis of SLE is still unclear, accumulating indications point to a role of lncRNAs in its development. Indeed, using gene expression profiles, Wang and colleagues identified 163 lncRNAs differentially expressed in SLE donors as compared with normal controls. Among them, 118 were upregulated and 45 downregulated in monocyte-derived dendritic cells (moDCs) which play important roles in the pathogenesis of SLE [95].

A previous study found that the lncRNA nuclear-enriched abundant transcript 1 (NEAT1) is associated with the pathogenesis of SLE [42]. NEAT1 expression was abnormally increased in SLE patients and predominantly expressed in human monocytes. Additionally, the expression of NEAT1 was induced by LPS through the activation of the p38 (mitogen-activated protein kinase) MAPK pathway. Silencing of NEAT1 in THP1 cells significantly reduced the expression of a group of chemokines and cytokines, including IL-6 and CXCL10, both of which were induced by LPS. Moreover, this study identified the involvement of NEAT1 in TLR4-mediated inflammatory process through the activation of the MAPK signaling pathway. Dysregulation of the TLR4 signaling was reported to be associated with the development of autoimmune diseases including SLE [96]. Importantly, there was a positive correlation between the expression level of NEAT1 and the clinical disease activity in SLE patients. Overall, overexpression of NEAT1 may contribute to the elevated production of cytokines and chemokines implicated SLE pathogenesis [42].

Xue and colleagues performed a transcriptome profiling of SLE patients and identified the linc00513 as a significantly overexpressed lincRNA in SLE. Interestingly, linc00513 overlaps with a functional SLE susceptibility locus in the promoter region, identified by genome-wide association studies, that resulted as a positive regulator of the IFN signaling pathway with consequent amplified IFN signaling in SLE patients. The identification of this susceptibility locus in the promoter region provides new understandings of the genetics of SLE predisposition and highlights the role of IFN signaling in SLE pathology [43,97]. Similarly, genome-wide association studies pointed to the chromosomal region 1q25 as an SLE-susceptible locus [98]. GAS5 (also involved in RA pathogenesis) is a prime candidate for the chromosome 1q25 SLE locus, suggesting a possible correlation between GAS5 alteration and SLE susceptibility [79]. Supporting this genetic evidence, it was shown that, in human SLE, the expression of GAS5 in plasma, as well as in CD4+ T cells and B cells, was decreased compared with healthy controls [44] and that GAS5 is involved in the apoptosis and growth arrest of human T cells, which are fundamental for the pathogenesis of SLE [99,100,101]. Together, these results provide evidence of the GAS5 role in SLE, suggesting its use in the diagnosis of the disease.

Using PBMCs collected from SLE patients and healthy controls, Cao and colleagues analyzed the expression of the lncRNA taurine-upregulated gene 1 (TUG1) to detect possible correlation with the pathological and clinical traits of SLE patients. Interestingly, the levels of the lncRNA TUG1 were significantly lower in the SLE patients, which was more evident in SLE patients affected by lupus nephritis. These results indicated that the lncRNA TUG1 may represent an interesting diagnostic tool for the identification of SLE patients with SLE patients with lupus nephritis providing new evidence for the diagnosis and prevention of SLE [45].

### 3.3. CircRNA and Systemic Lupus Erythematosus

Several studies have revealed that circRNAs are stably present in plasma and could act as biomarkers for diseases [102]. Unfortunately, studies on circRNA expression signatures in SLE are still very limited. In a recent study, 200 differentially-expressed circRNAs were identified in the plasma of SLE patients compared with that of normal controls. These circRNAs are generally distributed on all chromosomes, including the X chromosome with the upregulated circRNAs mainly encoded on chr6 (8.85%) and the downregulated circRNAs mostly found on chr16 (10.64%) [102].

Subsequently, Miao et al. have found 128 dysregulated circRNAs in patients with SLE compared with healthy controls [57]. Four circRNAs can be hypothetically used as biomarkers for SLE—two upregulated circRNAs (circLRRK2 and circPPHLN1) and two downregulated circRNAs (circPTPN22 and circMYBL1). circPTPN22, synthetized from the protein tyrosine phosphatase non-receptor type 22 (PTPN22) gene, potentially plays an important role in the pathogenesis of SLE and its expression is negatively correlated with disease activity. circPTPN22 acts as a “miRNA sponge”, binding and inactivating miRNA functions [57].

Li and colleagues analyzed the expression profile of circRNAs in children with SLE. Using circRNA microarray analysis, they compared the expression profiles of SLE patients with healthy controls and identified 348 circRNAs dysregulated—184 circRNAs upregulated and 164 downregulated. Additional analysis demonstrated that the levels of the circ_0057762 and circ_0003090 could be used for SLE diagnosis [58].

Two circRNAs, initially related to RA, were studied by Zhang and colleagues in the context of SLE by analyzing PBMCs from 18 diagnosed SLE patients and 10 healthy controls by RT-qPCR. The circRNAs studied, circ_0049224 and circ_0049220, target DNA methyltransferase 1 (DNMT1), and DNA hypomethylation has been correlated with SLE [59]. The results showed that the expression level of DNMT1 was positively linked to the expression of the two circRNAs and that these two circRNAs were downregulated in SLE patients [103]. Moreover, there was a negative correlation between the SLE activity and the expressions of circ_0049224 and circ_0049220 [103].

Studying by microarray the circRNAs expression profiles in T cells purified from patients with SLE and healthy controls, Li and colleagues identified 127 differentially-expressed circRNAs. Among those identified, decreased circ_0045272 expression was observed in T cells from SLE patients. Downregulation of the circ_0045272 in Jurkat cells induced early apoptosis and significantly enhanced IL-2 production [60], features of SLE onset.

Using next-generation sequencing (NGS) to study circRNA profiles in PBMCs from SLE patients, stratified by disease activity characteristics (stable or active SLE) and healthy controls, Guo and colleagues identified 84 circRNAs upregulated and 30 downregulated in patients with SLE compared with controls [104]. One of these, circ_0000479, showed a potential role as a biomarker for the diagnosis of SLE and, in a follow-up study, the authors suggested that the circ_0000479 levels in PBMCs and circ_0000479 combined with anti-dsDNA may serve as potential biomarkers for SLE diagnosis and evaluation of therapeutic effect [61].

These results may help to elucidate the pathogenesis of SLE and to identify new biomarkers and novel therapeutic strategies. However, future studies are needed to validate the results obtained until now and, thus, to clearly understand the role of the circRNAs in SLE.

## 4. Role of ncRNAs in Multiple Sclerosis

### 4.1. Multiple Sclerosis

MS, which affects more than 2 million people in the world, is the most prevalent chronic, autoimmune-mediated inflammatory disease of the central nervous system [105]. Many of the persons affected by MS develop impaired mobility, loss of sphincter control, and slowed cognitive processing [106]. Tissue damage in MS results from a complex and dynamic interaction between the immune system and neurons [107]. The immune response involves the participation of T helper (CD4+ T) and T cytotoxic (CD8+ T) lymphocytes, B lymphocytes, antibodies, and innate immune system cells [11,108,109]. The inflammatory response leads to demyelination and early neuronal transection [110]. As the disease progresses, a neurodegenerative process with more diffuse inflammation is observed [107,111]. The inflammatory and neurodegenerative processes can occur in parallel. Multifocal zones of inflammation due to T-lymphocytic and macrophage infiltrations, as well as oligodendrocyte death are the primary causes of myelin sheath destruction that leads to the formation of the characteristic central nervous system (CNS) plaques [12,112,113]. CNS plaques are composed of altered inflammatory cells and products (e.g., cytokines), demyelinated and transacted axons, and reactive astrocytosis, visible in both white and grey matter [114]. Subtypes of MS include relapsing remitting MS (RRMS), primary progressive MS (PPMS), secondary progressive MS (SPMS), and progressive relapsing MS (PRMS). RRMS is the most common subtype with approximately 87% of the diagnoses [106,115].

Current therapies for MS are mainly based on anti-inflammatory and immunomodulatory drugs. Unfortunately, no current treatments can halt the destruction of the nervous tissue and anti-inflammatory treatments are primarily effective in patients with RRMS, indeed only one treatment is today available for PPMS [116,117]. Additionally, the absence of valid and specific diagnostic and prognostic biomarkers for MS adds to the challenges in establishing efficient care for patients. Thus, new strategies are needed for early diagnosis and treatment of MS. In recent years, ncRNAs including miRNA, lncRNA, and circRNA have emerged as crucial players in different biological and physiological process including immunity and inflammation, thus these molecules were indicated to play significant roles in the pathogenesis of MS [118].

### 4.2. LncRNAs and Multiple Sclerosis

While the research about a causal role of lncRNAs in MS is just at the beginning, aberrant lncRNA expression has been reported in several studies. For example, screening for lncRNAs involved in autoimmunity and in the human inflammatory response identified three lncRNAs upregulated in the serum of RRMS patients, the NEAT1, TUG1, and 7SK small nuclear (RN7SK) RNA [46]. These lncRNAs play important roles in neurodegenerative processes indicating that these lncRNAs may be involved in MS pathogenesis [46]. Recently, Eftekharian et al. identified three circulating lncRNAs differentially expressed in blood cells of RRMS patients, compared to healthy controls, two downregulated lncRNAs (plasmacytoma variant translocation 1 (PVT1) and FAS antisense RNA 1 (FAS-AS1)) and one upregulated lncRNA (TNF and HNRNPL-related immunoregulatory long non-coding RNA (THRIL)) [47]. While the roles of lncRNAs in MS development remain unclear, their aberrant expression profiles suggest their candidacy as MS-specific biomarkers for disease diagnosis and progression as well as indicators for treatment response.

A primary cause and pivotal feature of MS pathogenesis is a pro-inflammation response and alteration in the cellular profile of high-M1 versus low-M2 polarized microglia [119]. Using microarray screening, Sun et al. identified the lncRNA, GAS5, as an epigenetic regulator of microglial polarization. GAS5 suppressed microglial M2 polarization and higher levels of GAS5 were found in amoeboid-shaped microglia in MS patients. Furthermore, performing functional studies they demonstrated that GAS5 suppressed transcription of TRF4, a key factor controlling M2 macrophage polarization, thereby inhibiting M2 polarization. Thus, GAS5 may be a promising target for the treatment of demyelinating diseases. [48].

Using lncRNA PCR arrays to analyze PBMCs from patients with MS, Fenoglio and colleagues identified a generalized dysregulation of lncRNA expression, including lncRNAs MALAT1 (metastasis-associated lung adenocarcinoma transcript 1), MEG9 (maternally-expressed 9), NRON (noncoding repressor of NFAT (nuclear factor of activated T cells)), ANRIL (CDKN2B antisense RNA 1), TUG1, XIST (X-inactive specific transcript), SOX2OT (SOX2 overlapping transcript), MIAT (myocardial infarction-associated transcript), HULC (hepatocellular carcinoma-associated transcript 1), and BACE-1AS (BACE1 antisense RNA). After experimental validation, decreased levels of NRON and TUG1 were confirmed in MS patients [49]. As we mentioned before, Santoro et al. showed an opposite trend of TUG1 expression, probably due to the different biological source chosen for the analysis or to a role of TUG1 in intercellular communications [46]. The lncRNA NRON interacts with members of the importin-beta superfamily and acts as a regulator of NFAT nuclear trafficking [120]. NFATs are transcription factors which regulate the transcription of genes coding for modulators of the immune system. Furthermore, NFAT contributes to oligodendroglial differentiation [121] thus alteration of NFAT activity by NRON may alter the immunological or pharmacological modulation of the myelination process during MS.

LncRNAs likely control the gene expression and functions of T helper Th1 and Th17 cells; these cells induce the secretion of cytokines leading to the typical MS immune-mediated inflammatory processes. Hosseini and colleagues identified lncRNAs in Th1 cells linked to MS by analyzing the expression levels of candidate lncRNAs and genes in 50 MS patients and 25 healthy controls. LncRNAs encoded by the gene AC007278.2 and IFNG antisense RNA 1 (IFNG-AS1-001) showed significantly higher expression in the relapsing MS phase whereas IFNG-AS1-003 was elevated in patients in the remitting MS phase compared with relapsing patients. Together, these deregulated lncRNAs may provide valuable tools for understanding the relationships between lncRNAs and MS progression as well as new pathological pathways [50]. Little is known about the precise roles of these lncRNAs but based on the expression profiles, lncRNAs appear to be important in the regulation of immune functions and are of potential biological significance for MS pathogenesis [122]. Although these findings are still preliminary, they are sufficient to identify an hypothesis for future investigation and give guidance regarding the design of future studies to analyze the role of lncRNA in MS onset, progression, and therapy.

### 4.3. CircRNA and Multiple Sclerosis

While accumulating evidence demonstrates key roles for circRNAs in the regulation of CNS and immune cell function, the functions of these RNAs in the pathogenesis of MS is still unknown and only few reports have addressed this topic. Over 400 circRNAs differentially expressed in the PBMCs of RRMS patients compared with those of healthy controls were initially identified by Cardamone and colleagues in 2017. They focused on the circ_0106803 which exhibited 2.8-fold upregulation expression in PBMCs of patients affected by RRMS, and was generated by alternative splicing abnormality of the Gasdermin B GSDMB gene [62]. The splicing variant is generally downregulated by the nonsense-mediated mRNA decay (NMD) in human cell lines, keeping the circ_0106803 levels significantly lower. Importantly, the identified circRNA is strongly dysregulated in PBMCs of RRMS donors, further supporting the concept that aberrant RNA metabolism is a feature of MS [62]. Subsequently, Cardamone has identified an upregulation of the lncRNA MALAT1 in MS patients. Alterations in MALAT1 expression have been implicated in alterations of back-splicing of circRNAs and splicing abnormalities of MS-associated genes such as IL7R and SP140 [123].

In a recent study, Paraboschi et al. [63] examined a circRNA (circ_0043813) derived from the signal transducer and activator of transcription 3 (STAT3), a gene involved in the inflammatory process and important for Th17 differentiation; circ_0043813 plays an important role in MS disease activity. Iparaguirre and colleagues have found that circ_0005402 and circ_0035560 were dysregulated in multiple sclerosis patients and could be used as biomarkers. These circRNAs are both located in chromosome 15 inside the ANXA2 gene and are both down-regulated in MS. The same circRNA expression trend was observed shortly after disease diagnosis in clinically-isolated syndrome patients [64]. Apart from the roles of circRNAs as novel contributors to the molecular pathways of immune dysregulation and CSN in MS, circRNAs may show great promise as novel biomarkers of MS disease.

## 5. Role of ncRNAs in Type 1 Diabetes Mellitus

### 5.1. Type 1 Diabetes Mellitus

T1D is an autoimmune condition that results from the destruction of β cells of the endocrine pancreas following infiltration of the islets of Langerhans by cells of the immune system (insulitis) [124]. This status leads to absolute insulin deficiency and stable hyperglycemia [124]. Genetic risk factors play a major role in T1D development, but the rapid increase in T1D incidence in recent decades demonstrates that environmental factors play a crucial role in its pathogenesis [125]. Additionally, epidemiological studies have suggested that a number of non-genetic factors, including diet, vitamin D, and gut microbiota, could play a role in T1D onset [126]. All of these factors, in a genetic-prone individual, could alter gene expression through epigenetic mechanisms inducing aberrant immune responses and islets autoimmunity [127]. For example, miRNAs have been found associated with insulin secretion and alteration of T1D risk [128].

In the pancreatic islets of patients with T1D, the majority of infiltrating immune cells represent CD8+ T cells while the remainder include mainly B cells, CD4+ T cells, and macrophages [129]. These infiltrating immune cells produce various pro-inflammatory cytokines which further exacerbate the T cell infiltration and apoptosis of β cells [130]. Animal studies and in vitro experiments support this T cell-mediated apoptosis as the main cause of death for the β cells [131,132]. Prior to T1D onset, a chronic atrophic inflammation within the islets of Langerhans is observed histologically, with the participation of T lymphocytes, macrophages, B lymphocytes, and dendritic cells [129]. This condition evolves over many years with symptomatic hyperglycemia occurring after a long latency period. The levels of pro-inflammatory cytokines and other specific markers, such as ncRNA, in the serum of the patients can be used as potential markers for T1D early diagnosis and prognosis [133].

### 5.2. LcnRNA and Type 1 Diabetes

Alterations in both the immune system and pancreatic β cells are the basis of T1D onset and pathogenesis [134]. In recent years, efforts to gain insights into the molecular mechanisms of pathogenesis of T1D have identified key roles for lncRNAs [135]. Indeed, using transcriptome profiling data of islets and β cells, more than 1000 islet-specific lncRNAs have been identified in human donors [136]. Downregulation of the trans-acting islet-specific lncRNA LINC01370 in mature β cells triggers the downregulation of the GLIS3 (GLIS family zinc finger 3) gene [51]. GLIS3 encodes an islet transcription factor and is a candidate gene for both type 1 and type 2 diabetes [137]. The nuclear-enriched β cell lncRNA PDX1 associated lncRNA, upregulator of transcription (PLUT) regulates the transcription of PDX1 (pancreatic and duodenal homeobox 1) mRNA. PDX1 is a key transcriptional regulator in the pancreatic β cells [52]. PLUT includes a cluster of enhancers that contact the PDX1 promoter in human islets regulating its transcription. Downregulation of PLUT correlated with reduced expression of PDX1 at both mRNA and protein levels in primary islet cells. Similar effects were observed for the mouse lncRNA orthologue MIN6, thus corroborating the role of PLUT [52]. Recently, another lncRNA-regulating PDX1 have been identified, MALAT1. The lncRNA MALAT1 leads to β cell dysfunction through reduction of the H3 histone acetylation of the PDX1 promoter [53]. Given the relevance of PDX1 in the differentiation of β-cells and in gene expression in the mature β-cell [52], inhibition of PDX1 by lncRNA may alter the number of healthful β-cells in adults [138]. Finally, the lncRNA TUG1, a highly conserved lncRNA, was studied in mouse pancreatic tissues. TUG1 was highly expressed in pancreatic tissue compared with other organ tissues, and expression was modulated by glucose. Furthermore, downregulation of TUG1 resulted in increased apoptosis and reduced insulin synthesis and secretion. These results propose a key role for TUG1 in β cell function and regulation and could consequently be involved in the onset of diabetes [54].

### 5.3. CircRNA and Type 1 Diabetes

Alterations in the circRNA expression profiles are also implicated in T1D. Indeed, it has been demonstrated that circRNAs are highly abundant in both α- and β-cells. Kaur and colleagues identified 10,830 circRNAs expressed in human α-, β-, and exocrine cells. The most highly expressed circRNAs were MAN1A2 (Mannosidase Alpha Class 1A Member 2), RMST (Rhabdomyosarcoma 2 Associated Transcript), and HIPK3 (Homeodomain Interacting Protein Kinase 3) across the three cell types. Interestingly, cell type-specific circRNAs in the three cell types as well as alternative circularization events were also observed. The circRNAs identified could serve as an important resource for future studies on β-cell function in relation to both type 1 and type 2 diabetes [65]. Importantly, specific cell-type gene expression in human pancreatic islets, including circRNA, is fundamental for the development of regeneration and therapeutic strategies to improve β-cell function in T1D.

The levels of two circular transcripts, circHIPK3 and circular RNA sponge for miR-7 / cerebellar degeneration-related protein 1 antisense transcript (ciRS-7/CDR1as) as, were found to be reduced in the islets of diabetic db/db mice. Forced downregulation of these transcripts in the islets of wild type animals resulted in reduced insulin secretion, β-cell proliferation, and survival. ciRS-7/CDR1as was previously proposed to function by sponging miR-7 and also regulating insulin secretion and β-cells proliferation. circHIPK3 acted by sequestering a group of miRNAs, such as miR-124-3p and miR-338-3p, and by regulating the expression of key β-cell genes, including SLC2A2 (solute-carrier family 2 member 2), AKT1 (AKT serine/threonine kinase 1), and MTPN (myotrophin) [66]. These preliminary findings identified circRNAs as key regulators of β-cell activities and suggest an involvement of this class of ncRNAs in β-cell dysfunction and T1D. Additional studies are needed to deeply clarify the role of circRNAs in this context.

## 6. Biomarkers

As mentioned above, the criteria currently used for autoimmune disease diagnosis are mainly based on clinical manifestations and laboratory tests [139]. Unfortunately, none of the markers used shows a high degree of specificity and sensitivity. Furthermore, autoimmune diseases such as SLE, exhibit various clinical manifestations [84], and other ADs, such as Sjogren’s syndrome and SLE, share similar symptoms and autoantibodies [140], making the diagnosis complicated. Therefore, the identification of new biomarkers with high sensitivity and high specificity for clinical therapy is a fundamental area of the modern biomedical research.

Biomarkers serve to facilitate disease diagnosis, reflecting disease activity, predicting future manifestations, and guiding drug administration [141]. LncRNAs, and circRNAs, in particular, are considered interesting biomarkers when they have specific criteria including quantitative alteration during a considered disease, tissue-specific expression indicating a specific pathogenic pathway, being easily accessible (e.g., circulating in blood), being stable in the considered body fluids, and being low cost [142]. In this manuscript, we have described several lncRNA that function mainly as biomarkers for ADs including the lncRNA HOTAIR [39] and ciRS-7 [80] for RA and TUG1 [45] and the circ_0000479 [61] for SLE. Unfortunately, until now none of the currently-identified possible biomarkers for ADs has all the desired characteristics [143] and been moved to clinical application. The field of biomarker discovery for ADs is still in its infancy compared to other areas such as cardiovascular disease and cancer [142,144,145]. Nevertheless, while ncRNA biomarker research is advancing for certain diseases, this field still faces several challenges related to preanalytical and analytical steps that strongly influence the results. Indeed, further studies are necessary to standardize the choice of the biological material (e.g., serum versus plasma) and the sample preparation protocols. Another key problem is related to the endogenous controls used for the analysis of lncRNAs and circRNAs in body fluids. Indeed, existing normalization approaches to reporting ncRNA levels are generally not standardized. Different reports use different normalization strategies, going from relative normalization with endogenous references to exogenous references [146]. Solid standardized methods for lncRNA isolation and normalization are fundamental to generating robust and reproducible results to facilitate the translation of lncRNA research into clinical application.

## 7. Susceptibility Loci and Long Noncoding RNA Annotations in Autoimmune Diseases

Genome-wide association studies (GWAS) have identified thousands of genetic variants associated with complex disease risk, including ADs. These variants often regulate quantitative traits such as gene expression levels (eQTLs, expression quantitative trait loci) and circulating immune cell levels, thus developing an extraordinary genetic characterization of the human genome. In keeping with other complex diseases, susceptibility loci for ADs are usually located in noncoding regions and even if many of them have been well characterized, the specific mechanism that mediated the disease risk is still largely unknown. Interestingly, several evidences suggest that GWAS susceptibility variants are enriched amongst linear and circular lncRNA [147]. Despite these studies, the pathological relevance of several lncRNAs in ADs remains unclear. Here, we decided to investigate the overlap between GWAS variants involved in AD susceptibility, and lncRNA annotations genetically linked by eQTLs. Our aim was to establish if AD-associated variants affect lncRNAs expression and whether this may contribute to the genetic mediation of disease risk.

In more detail, we extracted the genetic variants associated with ADs from the GWAS catalog (data release on 2020-03-08, converted in GRCh37), and searched them in the GTEx database (GTEx Analysis V6p and V7 releases), together with their proxies (r^2^ ≥ 80% in Europeans from the 1000 Genomes Project), focusing on eQTLs for antisense and lncRNA genes (Appendix A). Moreover, we also searched for circRNAs overlapping with protein coding eGenes (Appendix A).

When we analyzed the linear lncRNA family, we identified 32 specific lncRNAs for MS, 16 for SLE, 10 for RA and 7 for T1D. Among the most interesting results, the eQTL 8:11337811—highly correlated with RA and SLE susceptibility—modulates the expression of the antisense RP11-148O21.2, RP11-148O21.3, and RP11-148O21.4, and lncRNA modulates the expression of the antisense RP11-148O21.6 in whole blood. Although the correlation between the genes and the two ADs has been reported, their function, especially in the contest of RA and SLE, is still not understood.

In the context of MS, we identified several lncRNAs subject to genetic regulation, including the antisense lncRNAs CD27-AS1, which has been associated with MS by Disanto and colleagues, and HOXB-AS3, already related to cell proliferation and several forms of cancer [148]. Also, the MS-associated variant 11:118566746 (rs533646) influences lncRNA AP002954.4 expression. AP002954.4 is expressed specifically in the monocytic lineage of whole blood and according to Ricaño-Ponce and colleagues, this lncRNA can regulate cytokine responses [149]. Other MS-associated genetic variants regulate the expression of various lncRNAs, such as in the case of 12:6503500 (rs12296430), an eQTL for RP1-102E24.8 in various tissues (including whole blood, the spleen, and the nerve tibial) or 21:40463283 (rs9977672), an eQTL for AF064858.11 and AF064858.8 in whole blood. No functional roles in MS have been demonstrated so far for these lncRNAs, suggesting the need of further investigation.

Finally, analyzing genetic associations with T1D, we found that SNP 18:12779947 (rs2542151) is associated with increased risk of T1D and the expression of the lncRNA RP11-973H7.1 in pancreas tissue. A role of the lncRNA RP11-973H7.1 in T1D was already indicated by previous studies, however its function in pancreatic cells has not yet been studied [150].

Analyzing the circRNA family, we identified 832 circRNA specific for MS, 155 for RA, 388 for SLE, and 83 for T1D. Among them, the circ_0060573, which is transcribed by the CD40 gene, is shared among RA, SLE, and MS. No studies described this circRNA until now, but clarification of its role in AD pathogenesis could be relevant. Another interesting circRNA identified by our analysis is the circ_0022383 generated from the fatty acid desaturase 2 (FADS2) gene. Alteration in the circ_0022383 has been already demonstrated by Luo and colleagues during the analysis, by microarray, of circRNAs expression in peripheral blood in new-onset RA patients and healthy controls [151]. The specific function of the circ_0022383 still needs to be elucidated.

According to our results, the MS-associated variant 17:40529835 (rs1026916) influences the expression level of STAT3, overlapping with circ_0043813. Indeed, a key role of this circRNA in the pathogenesis of MS have been already demonstrated by Paraboschi and colleagues in 2018 [63]. The circ_0043813 derives from STAT3, known as a key gene for the signaling of pro-inflammatory cytokines [152] and Th17 cell differentiation, crucial event in MS pathogenesis [153]. Research on circRNAs is still at the beginning and additional studies are needed to clarify the role of circRNA in AD pathogenesis and to understand the genetic regulation. The simple overview performed here aims to describe the presence of linear and circular lncRNAs amongst GWAS susceptibility loci for ADs and the genetic relevance for these ncRNAs in the context of complex diseases such as autoimmune diseases.

## 8. Conclusions

Modern approaches to curing autoimmune diseases hold much promise, but they are based almost exclusively on targeting, activating, or inhibiting cellular protein factors that are involved in ADs. However, even if a majority of the human genome is transcribed into RNA, only around 1–2% of these transcripts encode for proteins. Consequently, most of the gene expression diversity in our cells is generated by the presence of ncRNAs that play important regulatory roles in all mammalian organs including the immune system [35,154]. As described above, lncRNAs including circRNAs can participate in the onset of complex diseases such as the ADs through different mechanisms modulating the expression levels of specific targets—chromatin modifications, splicing regulation, and miRNA sponging. Indeed, in the past years, several lines of evidence showed primary roles for lncRNAs in RA, SLE, MS, and T1D [23,30].

To date, the diagnosis of ADs is mainly based on clinical evidence. Unfortunately, in many patients, early symptoms of ADs can be nonspecific and suggestive of many other disorders. Imaging, such as magnetic resonance for MS, can assist in the diagnosis in certain cases, but a simple specific laboratory test will be much more convenient and affordable for identifying disease onset [155]. A contemporary priority of the ADs research field is to find specific biomarkers that will improve the clinical diagnosis while providing further insight into its pathophysiological mechanisms and possible new therapeutic approaches. As lncRNAs and circRNAs have been functionally connected to alterations of the immune response, it follows that alterations to ncRNA levels could be a reasonable way to detect the presence of autoimmune diseases in patients [156]. Recently, the identification of ncRNAs, such as circRNAs, in blood as well as other biological fluids has opened a new door for diagnosis of various diseases using samples obtained through non-invasive methods. Today the hope is that ncRNAs associated with autoimmune processes can be used to diagnose ADs before clinical manifestations appear. Additionally, whether lncRNAs can become therapeutic drug targets for the treatment of autoimmune diseases, and if ncRNAs can be used as diagnostic indicators of autoimmune disease still needs to be analyzed in detail.

GWAS have demonstrated their value as a central instrument for the identification of common genetic variants associated with complex traits and complex diseases, thus unravelling the genetic components of several pathological processes such as autoimmunity. Recently, with increased knowledge of the lncRNAs and circRNAs, thousands of studies have been initiated to characterize disease-associated variants situated in noncoding RNA regions. Additionally, several databases are trying to integrate genetic variants and ncRNAs with modern technologies to facilitate the identification of causal noncoding variants associated with disease. With our analysis, we wanted to highlight the relevance of noncoding variants in the context of the lncRNA family. Of course, specific research into the molecular genetics of lncRNA is necessary for a more complete understanding of their significance. ncRNA regulation in the context of autoimmune disorders and their implications in specific pathways will be solved in further studies relying on progressive next-generation sequencing and other modern molecular biologic techniques.

## Figures and Tables

**Figure 1 biomolecules-10-01044-f001:**
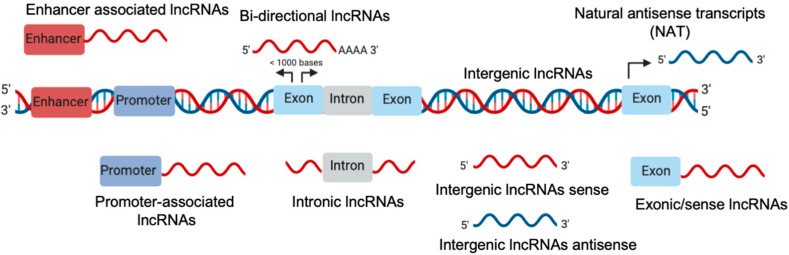
Schematic representation of long noncoding (lnc)RNA biogenesis.

**Figure 2 biomolecules-10-01044-f002:**
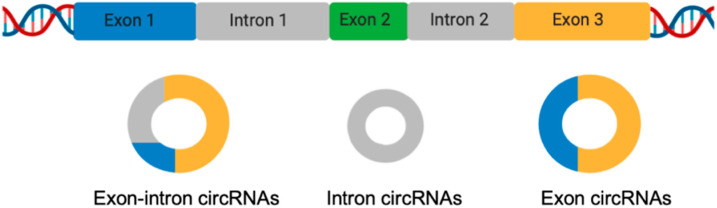
Schematic representation of circular RNA (circRNA) biogenesis.

**Table 1 biomolecules-10-01044-t001:** lncRNA in autoimmune diseases (ADs).

Disease	ncRNA Name	Regulation	Cellular Type/Biological Liquid	Ref.
**RA**	lncRNA AC000061	Upregulated	PBMCs	[37]
**RA**	HOTAIR	Upregulated	PBMCs/serum	[38]
**RA**	HOTAIR	Downregulated	PBMCs	[39]
**RA**	GAPLINC	Upregulated	FLSs	[28]
**RA**	ZFAS1	Upregulated	FLSs	[40]
**RA**	GAS5	Downregulated	CD4+T cells and B cells	[41]
**SLE**	NEAT1	Upregulated	Monocytes	[42]
**SLE**	Linc00513	Upregulated	PBMCs	[43]
**SLE**	GAS5	Downregulated	CD4+T cells and B cells/Plasma	[44]
**SLE**	TUG1	Downregulated	PBMCs	[45]
**MS**	NEAT1, TUG1, RN7SK	Upregulated	Serum	[46]
**MS**	PVT1, FAS-AS1	Downregulated	Blood	[47]
**MS**	THRIL	Upregulated	Blood	[47]
**MS**	GAS5	Upregulated	Primary cultured microglia	[48]
**MS**	MALAT1, MEG9, NRON, ANRIL, TUG1, XIST, SOX2OT, MIAT, HULC, BACE-1AS	Downregulated	PBMCs	[49]
**MS**	lncAC007278.2, IFNG-AS1-001, IFNG-AS1-003	Upregulated	PBMCs	[50]
**T1D**	LINC01370	Downregulated	Pancreatic islets, β cells	[51]
**T1D**	PLUT	Downregulated	Pancreatic β cells	[52]
**T1D**	MALAT1	Upregulated	Mouse islets and β cells	[53]
**T1D**	TUG1	Downregulated	Mouse pancreatic β cells	[54]

Legend: RA: Rheumatoid arthritis; SLE: Systemic lupus erythematosus; MS: Multiple sclerosis; T1D: Type 1 diabetes mellitus; PBMCs: Peripheral blood mononuclear cells; FLSs: Fibroblast-like synoviocytes; HOTAIR: HOX transcript antisense RNA; GAPLINC: gastric adenocarcinoma-associated, a positive CD44 regulator and long intergenic non-coding RNA; ZFAS1: ZNFX1 antisense RNA 1; GAS5: Growth-arrest-specific 5; NEAT1: Nuclear-enriched abundant transcript 1; TUG1: Taurine up-regulated gene 1; RN7SK: 7SK small nuclear; PVT1: Plasmacytoma variant translocation 1; FAS-AS1: FAS antisense RNA 1; THRIL: HNRNPL-related immunoregulatory long non-coding RNA; MALAT1: Metastasis-associated lung adenocarcinoma transcript 1; MEG9: Maternally-expressed 9; NRON: Noncoding repressor of NFAT (nuclear factor of activated T cells); ANRIL: CDKN2B antisense RNA 1; XIST: X-inactive specific transcript; SOX2OT: SOX2 overlapping transcript; MIAT: Myocardial infarction-associated transcript; HULC: Hepatocellular carcinoma-associated transcript 1; BACE-1AS: BACE1 antisense RNA; IFNG-AS1-001: IFNG antisense RNA 1; PLUT: PDX1 Associated lncRNA, Upregulator Of Transcription.

**Table 2 biomolecules-10-01044-t002:** circRNA in ADs

Disease	Noncoding (nc)RNA Name	Regulation	Cellular Type/Biological Liquid	Ref.
**RA**	circRNA_104194, circRNA_104593, circRNA_103334, circRNA_101407 circ_0092285, circ_0058794	Upregulated	PBMCs	[55]
**RA**	circRNA_102594 circ_0088088, circ_0038644	Downregulated	PBMCs	[55]
**RA**	circ_0008410,	Upregulated	PBMCs	[56]
**RA**	circ_0000175	Downregulated	PBMCs	[56]
**SLE**	circLRRK2 circPPHLN1	Upregulated	PBMCs	[57]
**SLE**	circPTPN22 circMYBL1	Downregulated	PBMCs	[57]
**SLE**	circ_0057762 circ_0003090	Upregulated	Blood	[58]
**SLE**	circ_0049224 circ_0049220	Downregulated	PBMCs	[59]
**SLE**	circ_0045272	Downregulated	T cells, Jurkat cells	[60]
**SLE**	circ_0000479	Upregulated	PBMCs	[61]
**MS**	circ_0106803	Upregulated	PBMCs	[62]
**MS**	circ_0043813	Upregulated	PBMCs	[63]
**MS**	circ_0005402 circ_0035560	Downregulated	PBMCs	[64]
**T1D**	MAN1A2, RMST, HIPK3	Upregulated	Human α-, β-, and exocrine cells	[65]
**T1D**	circHIPK3, ciRS-7/CDR1as	Downregulated	Human and mouse islets	[66]

Legend: RA: Rheumatoid arthritis; SLE: Systemic lupus erythematosus; MS: Multiple sclerosis; T1D: Type 1 diabetes mellitus; PBMCs: Peripheral blood mononuclear cells; circPTPN22: Protein tyrosine phosphatase non-receptor type 22; MAN1A2: Mannosidase Alpha Class 1A Member 2; RMST: Rhabdomyosarcoma 2 Associated Transcript; HIPK3: Homeodomain Interacting Protein Kinase 3; ciRS-7: Circular RNA sponge for miR-7/CDR1as: Cerebellar degeneration-related protein 1 antisense transcript).

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
