# Peer review of "Long Noncoding RNAs and Circular RNAs in Autoimmune Diseases"

_biomolecules, 2020, doi:10.3390/biom10071044_

Round 1
Reviewer 1 Report
This manuscript systematically review the progress of lncRNA research in multiple autoimmune diseases. Several concerns:
- It is good to see that circRNAs has been listed and reviewed. However, circRNAs belong to lncRNAs, I suggest in the description of lncRNAs section (i.e., "lncRNAs in autoimmune disease") to have some information of circRNAs.
- please keep the subtitles consistent, e.g., in lncRNAs the authors used RA, while in circRNAs the authors used rheumatoid arthritis.
- Very interesting in see the loci and lncRNAs section, wondering whether there is also information regarding the loci and circRNAs.
Author Response
Responses to the Comments from Reviewer 1:
It is good to see that circRNAs has been listed and reviewed. However, circRNAs belong to lncRNAs, I suggest in the description of lncRNAs section (i.e., "lncRNAs in autoimmune disease") to have some information of circRNAs.
[AU] We thank the reviewer for this request. Information’s regarding the circRNAs have been added in the lncRNAs section (line 76-79).
please keep the subtitles consistent, e.g., in lncRNAs the authors used RA, while in circRNAs the authors used rheumatoid arthritis.
[AU] We thank the reviewer for noticing it. Subtitles have been checked and corrected.
Very interesting in see the loci and lncRNAs section, wondering whether there is also information regarding the loci and circRNAs.
[AU] Following reviewer suggestion, we searched for loci overlapping with circRNAs and reported them in the Supplementary Table 1b. Also, we described the most interesting results in the related paragraph (lines 569-584).
Reviewer 2 Report
The paper by Lodde et al. reviewed the role of lncRNAs and circRNAs in four different autoimmune diseases. The review is well written and is of interest (despite a few reviews on this topic has been written in the last two years). Several points need to be reviewed by the authors.
- The introduction should describe the aim and structure of the review. Point 1.6 can be included in the introduction together with a brief description of autoimmune diseases, the rest can go in the main body. The review should be organized by diseases to simplify it.
As an example:
- Role of ncRNAs in rheumatoid arthritis
2.1 Rheumatoid arthritis (paragraph 1.2)
2.2 LncRNAs in RA (paragraph 2.1)
2.3 CircRNAs in RA (paragraph 3.1)
- Role of ncRNAs in SLE
3.1 SLE (paragraph 1.3)
3.2 LncRNAs in SLE (paragraph 2.2)
…
- In general the review lacks from some additional bibliography to sustain several sentences, in particular, in the first section referring to the description of the diseases. In addition, I miss some more critical discussion of the results and on the potential clinical applicability of the results.
- A section specific to the potential use of lncRNAs as biomarkers or therapy would be of special interest (even if other diseases may be also used as examples). I think that it is important for the reader to see the future potential use of lncRNAs in the clinical practice.
- Minor comments
-A figure with the biogenesis and the different types of lncRNAs (or even ncRNAs but with special attention to lncRNAs) can be of interest for the reader.
-Once an acronym is set, it has to be used throughout the manuscript [RA, miRNA, name of the miRNA (microR-X instead of miR-X)]. The name of some lncRNAs is fully shown (NEAT1, TUG1, ..) and other such as GAPLINC or ZFAS1 are not.
-Typo and grammar errors should be corrected.
-Sentence lines 230-232 should be rewritten.
-The authors have to explain why the location of GAS5 is susceptible to influence the development of SLE (lines 266-268).
Author Response
Responses to the Comments from Reviewer 2:
The introduction should describe the aim and structure of the review. Point 1.6 can be included in the introduction together with a brief description of autoimmune diseases, the rest can go in the main body. The review should be organized by diseases to simplify it.
As an example:
Role of ncRNAs in rheumatoid arthritis
2.1 Rheumatoid arthritis (paragraph 1.2)
2.2 LncRNAs in RA (paragraph 2.1)
2.3 CircRNAs in RA (paragraph 3.1)
Role of ncRNAs in SLE
3.1 SLE (paragraph 1.3)
3.2 LncRNAs in SLE (paragraph 2.2)
[AU] We appreciate the reviewer’s advice and requests. The revised manuscript has been reorganized according to reviewer’s suggestion. Thus, point 1.6, and introduction to points 2 and 3 have been included in the introduction. Additionally, the review has been organized by disease as suggested by the Reviewer 2.
In general the review lacks from some additional bibliography to sustain several sentences, in particular, in the first section referring to the description of the diseases.
[AU] We appreciate this concern. Additional references have been added to sustain sentences in the manuscript.
In addition, I miss some more critical discussion of the results and on the potential clinical applicability of the results.
[AU] We appreciate this concern. In the revised manuscript, we added discussion of the results and on the potential clinical applicability of the results (e.g. lines 144-148, 184-186, 276-277).
A section specific to the potential use of lncRNAs as biomarkers or therapy would be of special interest (even if other diseases may be also used as examples). I think that it is important for the reader to see the future potential use of lncRNAs in the clinical practice.
[AU] We appreciate this request. A section specific describing the potential use of lncRNAs as biomarkers has been added in the revised manuscript.
Minor comments
-A figure with the biogenesis and the different types of lncRNAs (or even ncRNAs but with special attention to lncRNAs) can be of interest for the reader.
[AU] We thank the reviewer for this request. In the revised manuscript, we have added two figures to describe biogenesis and classes of linear (Figure 1) and circular (Figure 2) lncRNA.
-Once an acronym is set, it has to be used throughout the manuscript [RA, miRNA, name of the miRNA (microR-X instead of miR-X)]. The name of some lncRNAs is fully shown (NEAT1, TUG1, ..) and other such as GAPLINC or ZFAS1 are not.
[AU] We appreciate the reviewer’s advice. Acronyms have been checked and corrected.
-Typo and grammar errors should be corrected.
[AU] We appreciate the reviewer’s suggestion. Type and grammar errors have been checked.
-Sentence lines 230-232 should be rewritten.
[AU] Following the Reviewer’s suggestions the sentence is now rephrased as follow: “Furthermore, using in vitro assays they demonstrated that ZFAS1 silencing reduce the migration and invasion of RA-FLS. Ye and colleague, finally, showed that ZFAS1 interacted with the miRNA-27a decreasing its functionality” (lines 176-178).
-The authors have to explain why the location of GAS5 is susceptible to influence the development of SLE (lines 266-268).
[AU] We thank the reviewer for this observation and apologize for our confusing description of this part. We corrected the sentence to describe that this region was implicated in SLE by genome-wide association studies and that the genetic evidence was supported by GAS5 described functions (lines 270-277).
Round 2
Reviewer 2 Report
The authors have answered to all comments. There is still a minor review that should be made by authors.
- There are still some diminutives that are set twice or not in the right place (lncRNAs: line 69 instead of line 61; circRNAs: line 69, line 86, and line 87; autoimmune disease: lines 106-107 instead of line 48; ncRNAs: line 134; STAT3 line 870 instead of line 870). Please check throughout the text.
- Please review grammar
- “Biomarkers” section must be numbered as 6. Please check numbers for the following sections
Author Response
To respond to the reviewer 2 comments, we have revised the text as follows:
-There are still some diminutives that are set twice or not in the right place (lncRNAs: line 69 instead of line 61; circRNAs: line 69, line 86, and line 87; autoimmune disease: lines 106-107 instead of line 48; ncRNAs: line 134; STAT3 line 870 instead of line 870). Please check throughout the text.
[AU] We thank the reviewer for this observation. Diminutives have been checked in the manuscript: lines 39-41, 52, 159, 224, 343, 414-419, 459, 500, 533,534
-Please review grammar
[AU] Type and grammar errors have been checked throughout the manuscript in lines 52, 60, 61, 63, 79, 82, 83, 104, 105, 106, 107, 142-144, 175, 265, 266, 330, 342, 343, 390, 546, 549, 559, 564, 572, 672.
-“Biomarkers” section must be numbered as 6. Please check numbers for the following sections. [AU] We thank the reviewer for noticing it. We revised the text as requested, line 539.